# Zearalenone Promotes Uterine Development of Weaned Gilts by Interfering with Serum Hormones and Up-Regulating Expression of Estrogen and Progesterone Receptors

**DOI:** 10.3390/toxins14110732

**Published:** 2022-10-26

**Authors:** Tingting Song, Xuemei Zhou, Xiangming Ma, Yanping Jiang, Weiren Yang, Faxiao Liu, Mei Liu, Libo Huang, Shuzhen Jiang

**Affiliations:** 1Key Laboratory of Efficient Utilization of Non-Grain Feed Resources (Co-Construction by Ministry and Province), Ministry of Agriculture and Rural Affairs, Department of Animal Science and Technology, Shandong Agricultural University, Tai’an 271018, China; 2Zhangjiakou Municipal Bureau of Agriculture and Rural Affairs, Zhangjiakou 075000, China; 3Dongying Science and Technology Innovation Service Center, No. 359 Nanyi Road, Dongying District, Dongying 257091, China; 4Zhongcheng Feed Technology Co., Ltd., Feicheng 271600, China

**Keywords:** zearalenone, gilts, uterus, estrogen receptors, progesterone receptor

## Abstract

In this study, we aimed to assess the effect of diet ZEA on serum hormones, the location and expression of estrogen receptor ERα/β and progesterone receptor (PR) of the uterus in weaned piglets and to reveal the mechanism underneath. A total of 40 healthy weaned gilts were randomly allocated to basal diet supplemented with 0 (Control), 0.5 (ZEA0.5), 1.0 (ZEA1.0) and 1.5 (ZEA1.5) mg ZEA/kg and fed individually for 35 days. Meanwhile, the porcine endometrial epithelial cells (PECs) were incubated for 24 h with ZEA at 0 (Control), 5 (ZEA5), 20 (ZEA20) and 80 (ZEA80) μmol/L, respectively. The results showed that nutrient apparent digestibility (CP and GE), nutrient apparent availability (ME/GE, BV and NPU), the uterine immunoreactive integrated optic density (IOD), relative mRNA and protein expression of ER-α, ER-β and PR and the relative mRNA and protein expression of ER-α and ER-β in PECs all increased linearly (*p* < 0.05) with ZEA. Collectively, ZEA can interfere with the secretion of some reproductive hormones in the serum and promote the expression of estrogen/progesterone receptors in the uterus and PECs. All these indicate that ZEA may promote the development of the uterus in weaned gilts through estrogen receptor pathway.

## 1. Introduction

According to the United Nations Food and Agriculture Organization, approximately 25% of food and feed output worldwide were contaminated with mycotoxins [1]; even the prevalence for the detected mycotoxins is up to 60–80%, according to recent research [2]. Zearalenone (ZEA), also known as F-2 toxin, is a nonsteroidal estrogenic mycotoxin produced by several strains of Fusarium species [3], which is stable and persists in the food and processed feed [4]. Studies have demonstrated that ZEA could increase the susceptibility of animals to infectious disease and induces severe lesions in the reproductive system [5] such as ovarian abnormalities, pseudopregnancy, infertility and miscarriage [6]. Additionally, ZEA also has cytotoxicity, immunotoxicity, carcinogenicity and genotoxicity [7,8], which poses a serious threat to animals’ health.

Estrogen is mainly secreted by ovaries and placenta, which plays an important role in promoting animal development and sexual organ maturation. Estrogen receptors (ERs) are cytoplasmic receptors that bind estrogen, migrate to the nucleus, regulate gene transcription and activate signal transduction pathways [9]. Similarly, the multiple biological effects of progesterone are mediated by interactions of this hormone with progesterone receptors (PR) [10]. It has been documented that ER and PR are commonly implicated in cell growth by hormone-induced cell proliferation [11]. However, ZEA has been reported to induce hormone disruption and toxic accumulation in reproductive organs, which might contribute to the abnormal expression of ERs [12,13]. The hypothalamic-pituitary-gonadal (HPG) axis plays an important role in reproductive endocrine function. It was demonstrated that endocrine disruptors could disturb the HPG axis and it adversely affected the function of the reproductive system [14]. Nevertheless, the estrogenic compound has the potency to bind with the nuclear ER throughout the HPG axis and impairs reproductive function by altering steroid hormone syntheses, such as estrogen and progesterone. Meanwhile, the variation of reproductive hormones might affect the HPG positive and negative feedback loop [15]. It is well known that the reproductive toxicity of ZEA is closely related to estrogen in vivo but the localization and expression of estrogen/progesterone receptors in the uteri of weaned gilts with low-dose ZEA in their diets are rarely reported. At the same time, the research on ZEA in vitro mainly involves ovarian granulosa cells and testicular sperm cells rather than porcine endometrial epithelial cells. Therefore, the main objective of this study was to investigate the effects of different concentrations of ZEA on serum reproductive hormones and hormone receptors (estrogen and progesterone receptors) of weaned gilts and to conduct in vitro verification through porcine endometrial epithelial cells, so as to explore the potential mechanism of ZEA in promoting uterine development.

## 2. Results

### 2.1. Nutrient Apparent Digestive and Metabolic Rate

The apparent digestibility (CP and GE) and the apparent metabolic rate (ME/GE, BV and NPU) of weaned gilts increased linearly (*p* < 0.05) with increasing dietary ZEA concentration (Table 1). The ME/GE of gilts of ZEA1.5 treatment was higher (*p* < 0.05) than that of the control and the BV and NPU of gilts by ZEA1.5 treatment was higher (*p* < 0.05) than that of the control and ZEA0.5 treatments.

### 2.2. Serum Hormone Level

Compared to the control, the piglets fed ZEA1.0 upregulated E_2_ and FSH (*p* < 0.05), while FSH and Pr were downregulated when fed ZEA1.5 (*p* < 0.05, Table 2).

### 2.3. The Localization of ER-α, ER-β and PR in the Uterus

The results of immunohistochemical analyses showed that in the uteri of the gilts, ER-α immunoreactive substance was mainly localized in the luminal epithelium cells (LE), glandular epithelial cells (G) and stromal cells (S) (Figure 1), while the ER-β and PR immunoreactive substance was mainly localized in the smooth muscle cells (M), luminal epithelium cells, glandular epithelial cells, stromal cells and vascular endothelial cells (V) (Figure 2 and Figure 3). Additionally, light yellow immunoreactive substances of ER-α, ER-β and PR were observed in the control group (A). No difference in the localization pattern of positive substance between ZEA and control was observed (the red arrows). Compared to the control, the positive reactions of ER-α in the luminal epithelium were enhanced (Figure 1(A2,B2,C2,D2)) but the reactions in lamina propria were weakened (Figure 1(A3,B3,C3,D3)) with increasing ZEA. However, the block localizations of yellow and brown immunoreactive substances of ER-β and PR were more obviously observed with increasing concentrations of ZEA (Figure 2(A2,B2,C2,D2) and Figure 3(A3,B3,C3,D3)). The IOD of ER-α, ER-β and PR in the uteri of weaned gilts increased linearly (*p* < 0.05) (Figure 4) and all of them in gilts fed ZEA1.5 were higher than those fed ZEA0.5 and control (*p* < 0.05).

### 2.4. The Relative mRNA Expression of ER-α, ER-β and PR in the Uterus

The linear increase in the relative mRNA expression of ER-α, ER-β and PR in the uteri of weaned gilts with increasing concentrations of ZEA was detected (*p* < 0.05) (Figure 4A). The relative mRNA expressions of ER-α, ER-β and PR in gilts fed ZEA1.5 and ZEA1.0 were higher than in those fed ZEA0.5 and the control (*p* < 0.05). Additionally, higher expression of ER-α was observed in piglets fed ZEA0.5 than those fed control (*p* < 0.05).

### 2.5. The Relative Protein Expression of ER-α, ER-β and PR in the Uterus

Western blot analysis revealed positive bands of appropriate sizes for all the studied genes (GAPDH, ER-α, ER-β and PR) (Figure 4B). The relative protein expressions of ER-α, ER-β and PR in the uteri of the weaned gilts increased linearly (*p* < 0.05) with the increased dietary ZEA concentration. Compared to the control, relative protein expressions of ER-α, ER-β and PR increased (*p* < 0.05) in gilts fed ZEA. Additionally, higher expressions of ER-α and PR were observed in piglets fed ZEA1.5 compared to those fed ZEA0.5 and ZEA1.0. The expressions of ER-β in piglets fed ZEA1.5 and ZEA1.0 were significantly higher than those fed ZEA0.5 (*p* < 0.05). There was a strong negative correlation between serum Pr and the protein expression of ER-α and PR. Meanwhile, a significant positive correlation between ER-α and PR in the uteri was detected in this study (*p* < 0.05).

### 2.6. The Relative mRNA and Protein Expression of ER-α and ER-β in Porcine Endometrial Epithelial Cells

The results of relative mRNA and protein expression of ER-α and ER-β in PECs were consistent with those in the uteri of the gilts (Figure 5). The relative mRNA and protein expression of ER-α and ER-β in PECs increased linearly (*p* < 0.05) with increasing concentrations of ZEA. The highest mRNA expressions of ER-α and ER-β were detected in PECs treated with ZEA80, followed by ZEA 20, ZEA5, and control (*p* < 0.05).

## 3. Discussion

The increasing awareness of animal feed safety issues due to feed contaminations by mycotoxins emphasizes the urgency of understanding the mechanism of toxin-induced animal diseases to find an applied approach for alleviating or eliminating toxins. In this study, we analyzed the effects of ZEA with high-purity, as commensal piglet in-feed toxin, on serum hormone and expressions of hormone receptors in weaned piglets. The similar feed intake between piglets fed ZEA with those fed the control indicated ZEA may not affect palatability [16].

One of the important findings of this study is the higher metabolic indicators of ME/GE, BV and NPU in piglets fed ZEA1.5, which may confirm the phenomenon of the anabolism of ZEA in the gastrointestinal tract [17]. The majority of previous studies have involved investigating the effect of ZEA on reproductive systems due to its estrogen activity and few studies were conducted on its impacts on the nutrient utilization of ZEA [18]. Jiang et al. (2012) reported that weaned gilts challenged with ZEA (1 mg/kg) had a significant reduction in apparent digestibility of nutrients (CP, Ca, P), BV and NPU [19]. A similar study demonstrated that the apparent digestibility of CP and DM of gilts decreased as the dietary ZEA (0, 0.2, 0.4 and 0.8 mg/kg) concentration increased [20]. However, Hauschild et al. (2007) reported that the nursery pigs fed a contaminated grain diet (ZEA, 2 mg/kg) could not affect the apparent digestibility of CP, DM and DE, which is inconsistent with our results [21]. Nevertheless, this study indicated that in-feed ZEA at different concentrations may have both beneficial (<1.0 mg/kg ZEA) and adverse influences (>1.0 mg/kg ZEA) on the nutrient absorption of piglets. Despite all, further investigation on the effects of ZEA with different concentrations of nutrient availability in piglets are necessary.

The anabolic effect of ZEA is accompanied by the interference with hormone levels in the body’s endocrine system because the ZEA has a special affinity with estradiol. In general, ZEA can cause serious hyperestrogenism in gilts [22,23]. However, the effects of ZEA on serum reproductive hormone levels were not consistent. FSH and LH play an important role in promoting the development of follicles and jointly promote estrogen secretion. In the present study, E2 and FSH were significantly increased in gilts with ZEA1.0 treatment. However, the concentration of LH did not change significantly. Wan et al. (2021) reported that ZEA could promote the expression of luteinizing hormone receptors (LHR) in the ovari of weaned gilts but no changes in LH were found in serum either [24]. In contrast, studies in prepubertal gilts (23.2 ± 0.68 kg) and piglets (8.19 ± 0.32 kg) found that ZEA significantly reduced serum LH levels [25,26,27]. The reason why ZEA has no effect on serum LH in our study may be explained by the corresponding insensitivity of serum LH to ZEA or the time dependence on ZEA. In addition, it is worth noting that the concentration of E2, Pr and FSH in serum decreased significantly in gilts with ZEA1.5 treatment. ZEA (0.25–1.04 mg/kg) was able to increase the level of E2 and FSH in the serum [24,25], as well as the same results in gilts with ZEA1.0 treatment. At the same time, more evidence pointed to the fact that high levels of ZEA (1.13–1.6 mg/kg) reduced the levels of reproductive hormone [25,28], which might be due to the inhibition of FSH synthesis of the pigs via the non-classical estrogen membrane receptor [29]. Another important reason is likely that excessive ZEA inhibits the synthesis and secretion of the follicle stimulating hormone via negative feedback [30], thus decreasing steroid production [29,31], in an attempt to maintain hormonal balance. However, the specific mechanism needs further study and it is now clear that weaned gilts fed ZEA-containing diets have disrupted reproductive hormone levels, which is likely to cause further damage to the body.

Estrogen receptors can only perform biological functions after being activated by estrogen [32]. In addition, the combination of estrogen and estrogen receptors can trigger rapid signaling pathways, such as the mitogen-activated protein kinase (MAPK) signaling pathway [33], thereby causing mitosis, anti-apoptosis, invasion and metastasis [34,35]. ER-α and ER-β are ligand-induced intracellular transcription factors that mediate many estrogen biological effects [36,37]. However, ZEA can only exert estrogen effects by combining with the ER-α and ER-β [38]. It was reported that ER-α was mainly distributed in endometrial cells and ovarian stromal cells, while ER-β was mainly distributed in ovarian granulosa cells, kidneys and bones [39]. The IOD of ER-α immunopositive substance in the uteri of the gilts was significantly higher than that of ER-β, which indicated that ER-α in the uterus was more extensive and played a major role in the present study. The results showed that the relative mRNA and protein expression of ER-α and ER-β in the uterus of the piglets and PECs increased linearly with increasing ZEA. However, there are different pathways and biological activities between ER-α and ER-β [40]. The ER-α mediates the differentiation and proliferation of epithelial cells, while ER-β mediates the apoptosis of cells [41]. Previous studies have shown that low-dose ZEA have no effect on ER-α in the uterus and ovaries [25,42]. However, the expression levels of ER genes were significantly affected by high doses of ZEA, which changed the HPG axis by altering gene expression of the steroid hormone-encoding gene to affect reproductive function [15]. Previous studies observed that long-term exposure to low doses of ZEA could alter ER-β genes and induce epigenetic modification to inhibit the development of the ovaries [43]. Likewise, our previous study showed that ZEA at 1.5 mg/kg was able to suppress the expression of GH and GHR, resulting in the suppression of ovarian development [44]. Although it did not inhibit uterine development, the high expression of ER-β suggested that high doses of ZEA may trigger apoptosis, which has also been demonstrated in much of the literature [22,45,46,47]. Therefore, we proposed that ZEA could promote uterine development by up-regulating the expression of ER-α but the high expression of ER-β suggested that ZEA might induce apoptosis while promoting cell proliferation. However, the mechanism of the above assumption needs to be validated in a following study.

The expression of progesterone depends on the PR. The combination of progesterone and PR could activate cytoplasmic kinase, causing PR to be modified, which plays a biological function by binding specific promoters [48]. At the same time, PR was a nuclear receptor with special affinity, which could regulate the expression of a large number of endometrial genes, thereby exerting an influence on the uterus [49]. Studies have found that estrogen could promote the expression of PR [50], which could induce the proliferation of stromal cells and glandular epithelial cells where they are expressed to help decidualization [51]. Our previous study showed that 0.5–1.5 mg ZEA/kg in the diet could up-regulate the expression of PR in the ovaries and regulate the development of follicles in the post-weaning piglets [16]. Consistent with the results of Yang et al. [16], as ZEA in the diet increased, the immunopositive reaction of PR in the uteri of piglets was enhanced and the relative expression of mRNA and protein of PR presented a linear increase. Therefore, ZEA (0.5–1.5 mg/kg) could induce high expression of PR in the uterus of the weaned gilts. However, it has been pointed out that PR can resist the over-proliferation of epithelial cells in the uterine cavity induced by estradiol [52]. Nevertheless, under the conditions of this experiment, the estrogen effect of ZEA seems to be more intense on the cells’ proliferation in the uterus, thereby promoting uterine development through a high expression of estrogen and progesterone receptors, then inducing uterine hypertrophy.

## 4. Conclusions

In summary, our results suggest that ZEA can induce serum hormonal disorders and accelerate uterine development by up-regulating the expression of estrogen/progesterone receptors in the uterus and PECs. These experimental data will help further understand the uterine hypertrophy induced by ZEA. However, it is expected that more target gene interference tests in vitro can further solve the reproductive diseases induced by ZEA.

## 5. Materials and Methods

### 5.1. Preparation of Zearalenone Diet

A 5-year survey program was initiated by our research group to evaluate the incidence of zearalenone in feed and feed raw materials in Shandong Province of China, including 1389 samples performed by the Asia Mycotoxin Analysis Center (Chaoyang University of Technology, Taichung, Taiwan). Results showed that the positive detection rate of ZEA reached 69.15%, the highest value of ZEA in compound feed samples was 4333.03 μg/kg and the average value of ZEA in compound feed samples was 972.56 μg/kg [53]. In addition, the Chinese standard of ZEA (2006) in weaning gilts is 0.5 mg/kg. Therefore, two (1.0 mg ZEA/kg) and three (1.5 mg ZEA/kg) times of the standard dose were used in this study [54,55]. Purified ZEA (Fermentek, Jerusalem, Israel) was dissolved in acetic ether and then poured onto talcum powder, which was left overnight to allow the acetic ether to evaporate. The ZEA premix of 1000 mg/kg was subsequently prepared, which was then diluted with toxin-free corn meal to prepare a premix containing 10 mg ZEA/kg. The diets were prepared in a bath and stored in separate covered containers before feeding.

### 5.2. Animals, Treatments and Feeding Management

The piglets in the experiment were raised in accordance with the guidelines for the care and use of laboratory animals provided by the Animals Nutrition Research Institute of Shandong Agricultural University and the Ministry of Agriculture of China. A total of 40 38-day-old healthy weaned gilts (Duroc × Landrace × Large White) with similar body weight (14.01 ± 0.86 kg) were selected in this study. The gilts were transferred to individual sterilized cages (0.48 m^2^) fitted with a plastic slatted floor, feed trough and nipple drinker. The temperature was set at 30 °C in week 1 and maintained at 26 °C to 28 °C for the rest of the trial with 65% relative humidity. The animals were randomly allocated to four dietary treatments (10 piglets per treatment as replicate) including basal diet (control) and basal diet with 0.5 (ZEA0.5), 1.0 (ZEA1.0) and 1.5 (ZEA1.5) mg ZEA/kg with analyzed concentrations of 0, 0.52 ± 0.07, 1.04 ± 0.03, and 1.51 ± 0.13 mg ZEA/kg, respectively. The basal diet (toxin-free) was formulated based on NRC (2012) and dietary treatments were applied to piglets for 35 days (the vulva size could be used as a clinical indicator for gilts fed with ZEA-contaminated diet, according to our previous study [24,55]) after 10-days adaptation (Table 3). The nutrient components of diets were sampled and analyzed at the beginning and end of the experiment.

### 5.3. Porcine Endometrial Epithelial Cells Culture and Treatment

The porcine endometrial epithelial cells (PECs) were kindly provided by the laboratory of metabolic diseases of Shandong Agricultural University [56]. The PECs were cultured with DMEM/F-12 (01-172-1ACS, Biological Industries, Kibbutz Beit Haemek, Israel) with 10% fetal bovine serum (FBS, FS101-02, TRAN, Beijing, China) in a cell incubator at 37 °C with 5% CO_2_.

The ZEA (Sigma, Z2125, Missouri, MO, USA) was dissolved in dimethyl sulfoxide (DMSO) (Sigma, D2650, Missouri, MO, USA) at the concentration of 20 mmol/L and stored at −20 °C. According to the cell activity (Appendix A), the cells were cultured and incubated in 6-well plates for 24 h with ZEA at 0 (Control), 5 (ZEA5), 20 (ZEA20) and 80 (ZEA80) μmol/L, respectively.

### 5.4. Sample Collection of Analyses of Nutrient Availability

Total collections of feces and urine were conducted in this study to determine the nutrient digestibility from d 15 of the experiment. Feces and urine from each piglet were weighted, pooled and collected in different bags daily. The feces and urine samples were stored at −20 °C. Crude protein (CP) and gross energy (GE) were analyzed based on AOAC (2012) and stored in hermetic containers. Approximately 2.5% of daily feces and 1% of daily urine excretions were preserved in 1:3 diluted acid (250 mL concentrated sulfuric acid per liter solution) for analyzing nitrogen contents (CP = Nitrogen × 6.25). Subsamples of feed, urine, and feces were dried in a 65 °C drying oven for 48 h, finely ground with a pulverizer, and stored in hermetic containers for GE and CP analyses (AOAC, 2012).

The nitrogen contents were used to calculate net protein utilization (NPU) and protein biological value (BV):NPU = (RN/IN) × 100;
BV = (RN/DN) × 100
where DN (digested N) = ingested N-fecal N; RN (retained N) = ingested N-fecal N-urinary N.

The gross energy of feed, urine and feces was used to calculate digestive energy (DE) and metabolizable energy (ME):Digestive energy (DE) = GE − fecal energy;
Metabolic energy (ME) = GE − fecal energy − urinary energy.

### 5.5. Sample Collection of Blood and Uterus

On d 35 of the trial, vacuum anticoagulation tubes (with K2EDTA) were used to collect blood samples from the jugular vein of all piglets (15 mL per piglet). All gilts were fasted for 12 h before the collection and collected blood samples were incubated (37 °C for 2 h) and then centrifuged (1500× *g* for 10 min). The serum after centrifugation was stored in 1.5 mL Eppendorf tubes at −20 °C for hormone analysis. All piglets were euthanized by electrocution (head only, 110 V, 60 Hz) after blood collection and two uterine samples were immediately excised sterilely. One uterine was placed in an RNase-free 2-mL frozen tube and stored at −80 °C for subsequent analysis. The other sample was promptly fixed in Bouin’s solution for immunohistochemical analysis. After fixation, 5-μm-thick sections were cut using a Leica RM 2235 microtome (Lecia, Nussloch, Germany), mounted on poly L-lysine-coated glass slides, and dried overnight at 37 °C before routine staining for immunohistochemical analysis.

### 5.6. Determination of Serum Hormones

Serum content of estradiol (E_2_), progesterone (P), follicle stimulating hormone (FSH) and luteinizing hormone (LH) was analyzed by radioimmunoassay (RIA) using commercial RIA Kit (Nanjing Jiancheng Bioengineering Institute, Nanjing, China) according to the manufacturer’s protocol with SN-682RIA gamma counter (Shanghai Fuguang nuclear Photoelectric Instrument Co., Ltd., Shanghai, China). Hormone levels per gilt were determined in triplicate to avoid inter-assay variation.

### 5.7. Immunohistochemical Analysis and Integrated Optical Density of ER-α, ER-β and PR

The histological sections of the uterus were dewaxed and rehydrated for immunohistochemical chemical analysis according to standard immunohistochemistry protocols [55,56,57]. Antigen retrieval was performed in sodium citrate buffer (0.01 mol/L, pH 6.0) using a microwave unit for 20 min at full power. The sections were then washed with phosphate-buffered saline (PBS) (0.01 mol/L, pH 7.2).

Five stained sections randomly selected from ten gilts in each group were examined under a microscope (ELIPSE 80i; Nikon, Tokyo, Japan). The amount of cell staining and quantity of the target antigen of ER-α, ER-β and PR were evaluated and analyzed by Image Pro-Plus 6.0 (Media Cybernetics, Silver Spring, MD, USA). This yielded value of the total cross-sectional integrated optical density (IOD) [57] was used to compare the staining amount in the different treatments.

### 5.8. Quantification mRNA Expression Using Quantitative Real-Time Polymerase Chain Reaction

Total RNA was extracted from the uteri of gilts and PECs using RNAiso Plus (Applied TaKaRa, Dalian, China) according to the manufacturer’s instructions. The RNA purity was checked by an Eppendorf Biophotometer (RS323C; Eppendorf, Leipzig, Germany) and cDNA was made by a Reverse Transcription System kit (PrimeScript RT Master Mix, RR036A; Applied TaKaRa).

Quantitative Real-time PCR (RT-PCR) of genes including ER-α, ER-β and PR were determined by SYBR Premix Ex Taq-TIi RNaseH Plus (code: RR420A, Lot: AK7502; Applied TaKaRa) in an AB7500 real-time PCR system (Applied Biosystems, California, CA, USA). The cycling conditions were 95 °C for 30 s, followed by 43 cycles at 95 °C for 5 s, 60 °C for 34 s, 95 °C for 15 s, and 60 °C for 60 s, with a final step at 95 °C for 5 s. The relative gene expressions were calculated using the 2^−ΔΔ^CT method [58]. Each sample was determined in triplicate to avoid inter-assay variation. The primer sequences and product lengths are presented in Table 4.

### 5.9. Western Blot Analysis

Total protein was extracted from the uterus and the PECs using radioimmunoprecipitation assay lysis buffer supplemented with phenylmethanesulfonyl fluoride (Beyotime Biotechnology, Shanghai, China) according to the manufacturer’s instructions and detected by the Bicinchoninic Acid Protein Assay Kit (Tiangen Biotech Co., Ltd., Beijing, China). Approximately 50 μg of protein was separated by electrophoresis on polyacrylamide gels and subsequently transferred to nitrocellulose membranes. The membranes were blocked with 5% skim milk powder for 1.5 h and incubated overnight at 4 °C with the following primary antibodies: anti-ER-α (1:1000; ab3575, Abcam, Shanghai, China), anti- ER-β (1:1000; ab3576, Abcam), anti-PR (1:200; sc-539, Santa Cruz Biotechnology, Shanghai, China), anti-GAPDH (1:5000; Abcam) and anti-actin (1:5000; Abcam). After incubation, the membranes were washed three times with TBST and incubated with anti-rabbit/mouse IgG antibody (1:2000; CWBIO, Beijing, China) for 2 h at 37 °C, followed by washing with TBST. Finally, the membranes were immersed in a high-sensitivity luminescence reagent (BeyoECL Plus; Beyotime), exposed to film using a FusionCapt Advance FX7 (Beijing Oriental Science and Technology Development Co., Ltd., Beijing, China) and analyzed using Ipp 6.0 (Image Pro-Plus 6.0; Media Cybernetics).

### 5.10. Statistical and Analysis

The experiment was evaluated as a complete random design (CRD) and each piglet was considered an experimental unit. All data were analyzed using one-way analysis of variance (ANOVA) by general linear model (PROC GLM) with the model (SAS 9.2): The sample *Yij = µ + Ti + eij*, where *µ* is the total means, *Ti* is the fixed treatment effects (*i* = 1, 2, 3, 4), *eij* is the residual of the model. Orthogonal polynomial contrasts were then used to determine linear responses to dietary ZEA concentrations. Significant differences were determined using Duncan’s multiple-range tests. Statistical significance was considered at *p* < 0.05.

## Figures and Tables

**Figure 1 toxins-14-00732-f001:**
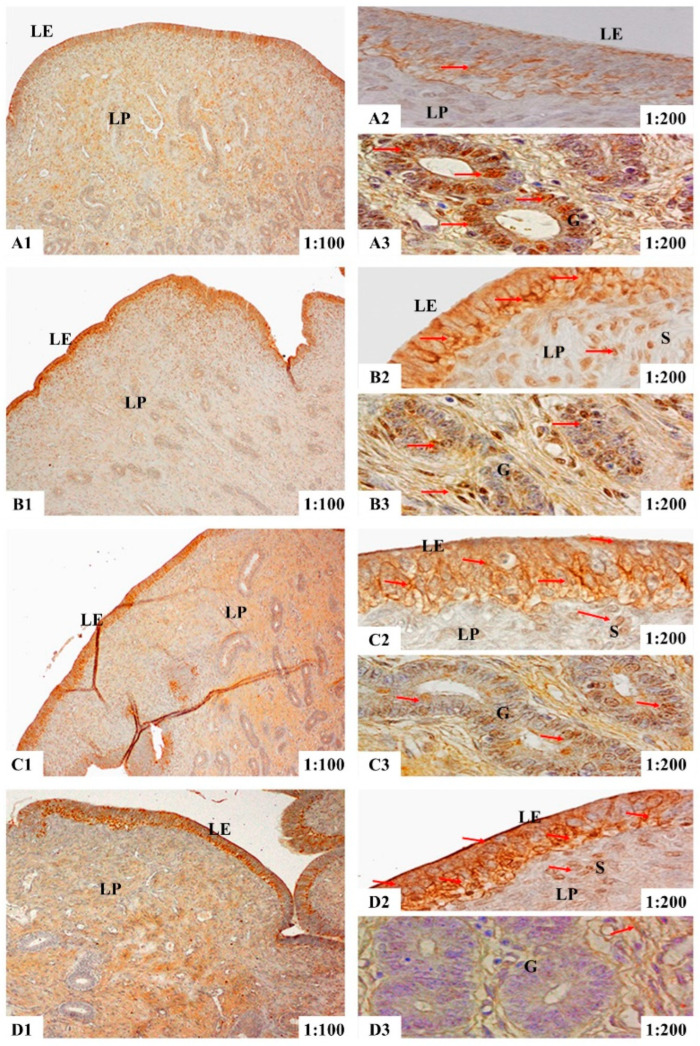
Effects of zearalenone (ZEA) on the localization of estrogen receptor α (ER-α) in uteri of weaned gilts. Control (**A**), ZEA0.5 (**B**), ZEA1.0 (**C**) and ZEA1.5 (**D**) represent the basal diet supplemented with 0, 0.5, 1.0 and 1.5 mg ZEA/kg. **A1**, **B1**, **C1** and **D1** are the uterus. More specifically, **A2**, **B2**, **C2** and **D2** are the luminal epithelium, **A3**, **B3**, **C3** and **D3** are the lamina propria. The 1:100 and 1:200 represent the 100× and 200× sample views, respectively. The red arrows indicate the immunoreactivity. The LE is luminal epithelium, LP is lamina propria, G is uterine gland and S is stromal cells.

**Figure 2 toxins-14-00732-f002:**
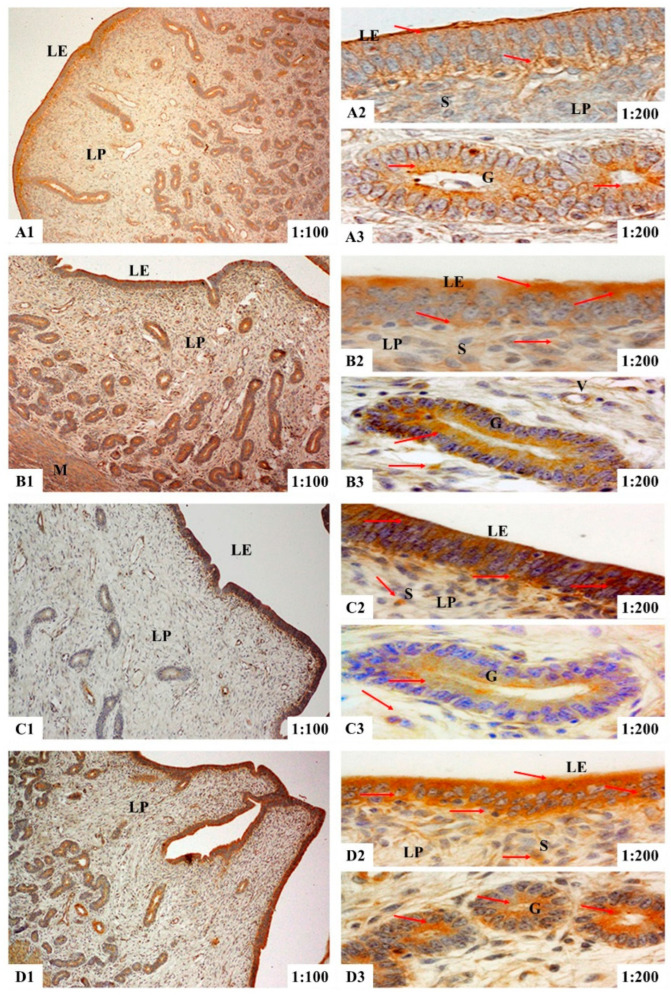
Effects of zearalenone (ZEA) on the localization of estrogen receptor β (ER-β) in uteri of weaned gilts. Control (**A**), ZEA0.5 (**B**), ZEA1.0 (**C**) and ZEA1.5 (**D**) represent the basal diet supplemented with 0, 0.5, 1.0 and 1.5 mg ZEA/kg. **A1**, **B1**, **C1** and **D1** are the uterus. More specifically, **A2**, **B2**, **C2** and **D2** are the luminal epithelium, **A3**, **B3**, **C3** and **D3** are the lamina propria. The 1:100 and 1:200 represent the 100× and 200× sample views, respectively. The red arrows indicate the immunoreactivity. The M is myometrium, LE is luminal epithelium, LP is lamina propria, G is uterine gland, S is stromal cells and V is vessel.

**Figure 3 toxins-14-00732-f003:**
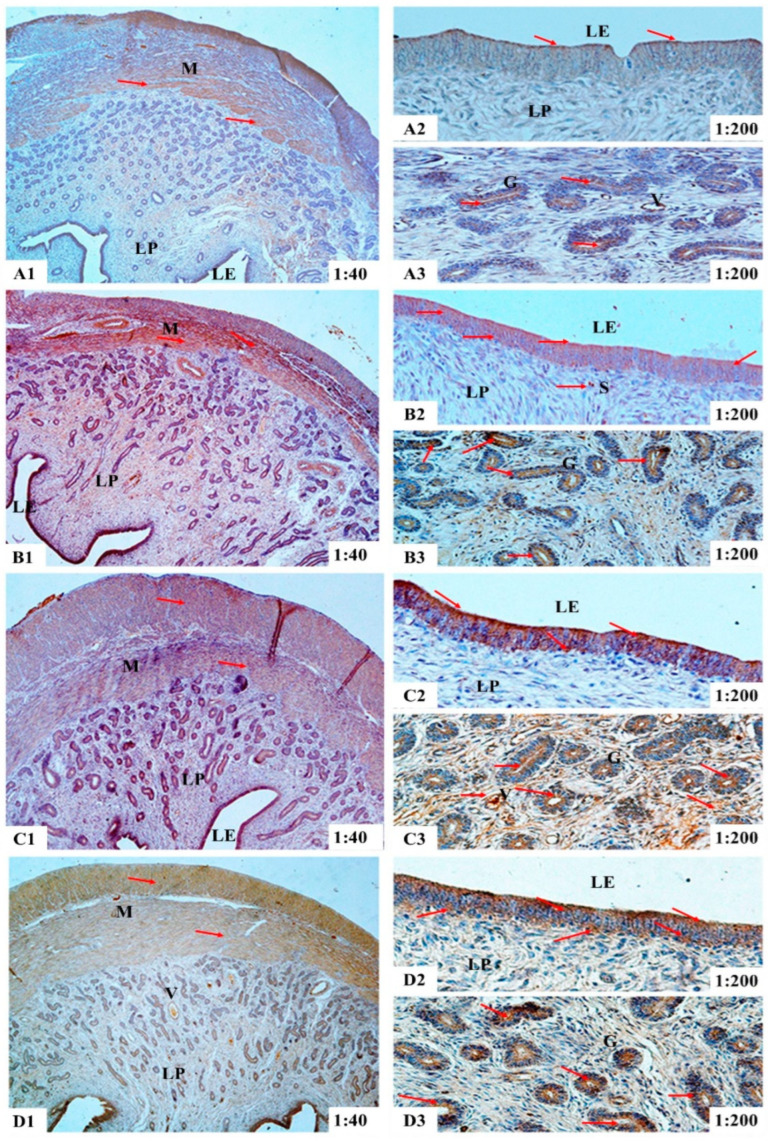
Effects of zearalenone (ZEA) on the localization of progesterone receptor (PR) in uteri of weaned gilts. Control (**A**), ZEA0.5 (**B**), ZEA1.0 (**C**) and ZEA1.5 (**D**) represent the basal diet supplemented with 0, 0.5, 1.0 and 1.5 mg ZEA/kg. **A1**, **B1**, **C1** and **D1** are the uterus. More specifically, **A2**, **B2**, **C2** and **D2** are the luminal epithelium, **A3**, **B3**, **C3** and **D3** are the lamina propria. The 1:40 and 1:200 represent the 40× and 200× sample views, respectively. The red arrows indicate the immunoreactivity. The M is myometrium, LE is luminal epithelium, LP is lamina propria, G is uterine gland, S is stromal cells and V is vessel.

**Figure 4 toxins-14-00732-f004:**
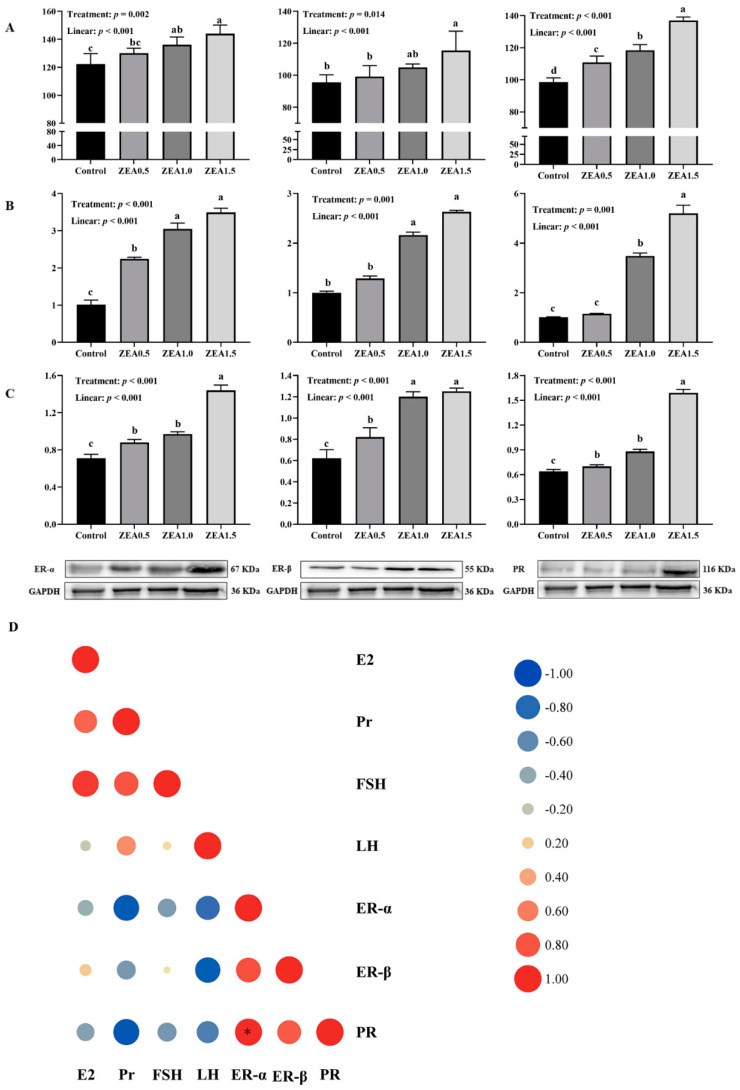
Effects of zearalenone on the immunoreactive integrated optic density (IOD, (**A**)) relative mRNA (**B**) and protein (**C**) expression of the estrogen receptor α (ER-α), estrogen receptor β (ER-β) and progesterone receptor (PR) in uteri of weaned gilts. (**D**) The Pearson correlation between serum hormones and estrogen receptors and progesterone receptors. Circles in the balloon plot denoting the direction of the association are colored red (positive) and blue (negative). Control, ZEA0.5, ZEA1.0 and ZEA1.5 represent the basal diet supplemented with 0, 0.5, 1.0 and 1.5 mg ZEA/kg, respectively. Different letters on the column indicate significant differences (*p* < 0.05). Different letters (a, b, c and d) and “*” represent significant correlations (*p* < 0.05).

**Figure 5 toxins-14-00732-f005:**
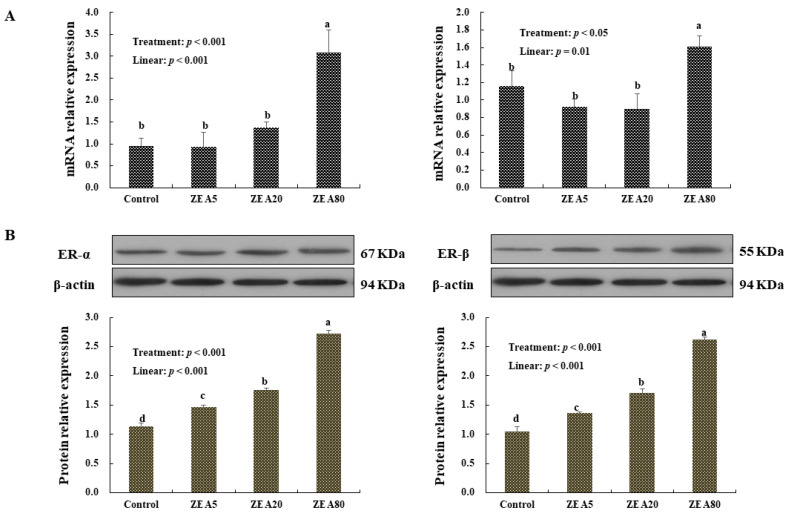
Effects of zearalenone on the relative mRNA (**A**) and protein (**B**) expressions of the estrogen receptor α (ER-α) and estrogen receptor β (ER-β) of the porcine endometrial epithelial cells (PECs) exposed to ZEA at 0 (Control), 5 (ZEA5), 20 (ZEA20), and 80 (ZEA80) μmol/L for 24 h. Different letters (a, b, c and d) on the column indicate significant differences (*p* < 0.05).

**Table 1 toxins-14-00732-t001:** Effect of zearalenone (ZEA) on apparent digestibility and availability of nutrients in weaned gilts (n = 10).

Items	Control	ZEA0.5	ZEA1.0	ZEA1.5	SEM	*p*-Value	
						Treatment	Linear
Apparent digestibility, %							
CP	84.80	84.78	85.10	85.69	0.433	0.391	0.043
GE	85.82	86.43	86.71	86.93	0.451	0.429	0.045
Apparent metabolic rate, %							
ME/GE	76.54 ^b^	77.51 ^ab^	78.04 ^ab^	79.61 ^a^	0.777	0.041	0.017
BV	66.06 ^b^	66.57 ^b^	67.55 ^ab^	68.12 ^a^	0.781	0.048	0.024
NPU	60.11 ^b^	60.59 ^b^	61.46 ^ab^	62.14 ^a^	0.552	0.039	0.026

Control, ZEA0.5, ZEA1.0 and ZEA1.5 represent the basal diet supplemented with 0, 0.5, 1.0 and 1.5 mg ZEA/kg ZEA and with analyzed ZEA concentrations of 0, 0.52 ± 0.07, 1.04 ± 0.03 and 1.51 ± 0.13 mg/kg, respectively. Data per piglet were run in duplicate in a single assay to avoid inter-assay variation. SEM, standard error of the means. Values in the same line with different letters mean significantly different (*p* < 0.05).

**Table 2 toxins-14-00732-t002:** Effects of zearalenone (ZEA) on serum hormones of weaned gilts (n = 10).

Items	Control	ZEA0.5	ZEA1.0	ZEA1.5	SEM	*p*-Value	
						Treatment	Linear
E_2_, ng/mL	11.89 ^bc^	12.74 ^b^	14.65 ^a^	11.15 ^c^	0.644	<0.001	0.921
Pr, ng/mL	1.03 ^a^	0.99 ^a^	1.06 ^a^	0.75 ^b^	0.012	<0.001	0.101
FSH, mIU/mL	2.86 ^b^	2.52 ^b^	4.12 ^a^	1.89 ^c^	0.074	<0.001	0.440
LH, mIU/mL	3.09	3.00	2.99	2.97	0.051	0.825	0.380

E_2_, Estradiol; Pr, Progesterone; FSH, Follicle stimulation hormone; LH, Luteinizing hormone. Control, ZEA0.5, ZEA1.0 and ZEA1.5 represent the basal diet supplemented with 0, 0.5, 1.0 and 1.5 mg ZEA/kg and with analyzed ZEA concentrations of 0, 0.52 ± 0.07, 1.04 ± 0.03 and 1.51 ± 0.13 mg/kg, respectively. Data per piglet were run in duplicate in a single assay to avoid inter-assay variation. SEM, standard error of the means. Values in the same line with different letters mean significantly different (*p* < 0.05).

**Table 3 toxins-14-00732-t003:** Ingredients and nutrient levels of the basal diet (air dry basis) ^1^.

Ingredients	Content (%)	Nutrients ^3^	
Corn	64.5	Digestible Energy, MJ/kg	13.81
Whey powder	5.0	Crude Protein (%)	19.82
Soybean meal	23.0	Calcium (%)	0.70
Fish meal	5.0	Total Phosphorus (%)	0.64
L-Lysine HCl	0.2	Lysine (%)	1.22
CaHPO4	0.7	Sulfur Amino Acid (%)	0.65
Pulverized Limestone	0.3	Threonine (%)	0.75
NaCl	0.3	Tryptophan (%)	0.22
Premix ^2^	1.0		
Total	100.0		

^1^ Treatments were basal diets supplemented with 0, 0.5, 1.0 or 1.5 mg ZEA/kg, with analyzed ZEA concentrations of 0, 0.52 ± 0.07, 1.04 ± 0.03 and 1.51 ± 0.13 mg/kg, respectively. ^2^ Supplied per kg of diet: VA 3300 IU, VD3 330 IU, VE 24 IU, VK3 0.75 mg, VB1 1.50 mg, VB2 5.25 mg, VB12 0.026 mg, pantothenic acid 15.00 mg, niacin 22.50 mg, biotin 0.075 mg, folic acid 0.45 mg, Mn (MnSO4·H2O) 6.00 mg, Fe (FeSO4·H2O) 150 mg, Zn (ZnSO4·H2O) 150 mg, Cu (CuSO4·5H2O) 9.00 mg, I (KIO3) 0.21 mg, Se (Na2SeO3) 0.45 mg.^3^ The digestible energy was the calculated value of the digestion test and the other nutrient levels were the analyzed value.

**Table 4 toxins-14-00732-t004:** Primer sequences of glyceraldehyde-3-phosphate (GAPDH), estrogen receptors (ERs) and progesterone receptor (PR).

Target Gene	Accession	Primer Sequence (5′ to 3′)	Product Size, bp
GAPDH	NM-001206359.1	F: ATGGTGAAGGTCGGAGTGAA	154
		R: CGTGGGTGGAATCATACTGG	
ER-α	NM-214220.1	F: GACAGGAACCAGGGCAAGT	125
		R: ATGATGGATTTGAGGCACAC	
ER-β	NM-001001533.1	F: ATGCCTTGGTCTGGGTGAT	120
		R: GTTCCGTGCCCTTGTTACTG	
PR	NC-010451.3	F: ATGAAAGCCAAGCCCTAAGC	180
		R: GGCAGTGACTTAGACCACGC	

## Data Availability

All data generated or analyzed during this study are included in this published article.

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
