# Peer review of "Zearalenone Promotes Uterine Development of Weaned Gilts by Interfering with Serum Hormones and Up-Regulating Expression of Estrogen and Progesterone Receptors"

_toxins, 2022, doi:10.3390/toxins14110732_

Round 1
Reviewer 1 Report
General thoughts
1. Why was the dose calculated per kg of feed and not per kg BW (the latter would probably be more appropriate as each gilt always receives the same amount). So the degree of exposure depends on the prey of the particular animal (gilt). I know from my own experience that it is different.
2. Throughout the work, I propose to change the dose amount, e.g. 1.5 mg ZEA / kg of feed - it would be more legible.
3. Why were gilts slaughtered after 35 days of exposure to mycotoxin?
4. No justification was made for the selection of ZEA exposure doses administered via feed. Referring to other authors of scientific studies is not enough.
5. There is also no rationale for selecting ZEA doses during the 24-hour culture of porcine endometrial cells.
6. There is no justification as to whether there is any correlation between the ZEA exposure doses (contaminants) in the feed and the exposure values during culturing the endometrial epithelial cells of gilts.
7. The biotransformation processes of zearalenone at the intestinal level and in the blood of the gilts during the 35-day intoxication period were completely forgotten. About adaptive processes, especially adaptive immunity, which begins around the 7th day of exposure to ZEA and its metabolites in the organism of matured gilts. The biological activity of alpha-zearalenol is much greater than that of the parent substance.
8. There is no compatibility between the title, purpose and conclusions.
9. There is a provision for uterine development in the abstract and conclusion. In my opinion, there is degeneration and atrophy of celles and tissues, particularly in endometrium and myometrium and extensive edema and numerous extravasations were typical of estrogen-induced lesions. Therefore, it is not development, but a disease state (pathology) accompanying zearalenone mycotoxicosis.
Detailed comments
Lines 40-43 - The interaction of ZEA with steroid hormones, ERs, PRs and the hypothalamic-pituitary-gonad axis has been forgotten;
Line 155 - What does purity mean - were the presence of ZAE metabolites checked, for example?
Line 155 - I think there should be another term. Commensalism is a long-term biological interaction (symbiosis) in which members of one species gain benefits while those of the other species neither benefit nor are harmed. It is an undesirable substance, i.e. a substance that is present in the environment without human intervention but that may be dangerous to mammals.
Lines 183-185 - It is difficult to relate the results of pregnant sows to the results of maturing gilts
Author Response
Dear reviewer 1:
On behalf of my co-authors, we thank you very much for giving us an opportunity to revise our manuscript (toxins-1947616), we appreciate you very much for their positive and constructive comments and suggestions on our manuscript. Those comments are all valuable and very helpful for revising and improving our researches. We have studied the valuable comments carefully and have made correction which we hope meet with approval.
Reviewer 1:
General thoughts:
- Why was the dose calculated per kg of feed and not per kg BW (the latter would probably be more appropriate as each gilt always receives the same amount). So the degree of exposure depends on the prey of the particular animal (gilt). I know from my own experience that it is different.
-- As it is described in the “Materials and Methods (L 278)”, the gilts have similar body weight (14.01 ± 0.85 kg). The piglets were allowed ad libitum to feed and water. ZEA had no significant effect on the average daily gain, average daily feed intake and feed to weight ratio of weaned gilts (Yang et al., 2017). Therefore, the amount of ingested ZEA by each pig per kg BW was the similar.
Yang, L., Wang, S., Yang, W., Huang, L., Liu, F., Jiang, S., Yang Z. (2017). Effects of zearalenone on production performance, serum antioxidant capacity and immune function of weaning gilts. Chinese Journal of Animal Nutrition. 29 (08): 2843-2850.
- Throughout the work, I propose to change the dose amount, e.g. 1.5 mg ZEA / kg of feed - it would be more legible
-- The dose amount “mg / kg” was changed to “mg ZEA / kg”, based on your suggestion.
- Why were gilts slaughtered after 35 days of exposure to mycotoxin?
The vulva size can be used as a clinical indicator of gilts fed ZEA-contaminated diet according to our literature (Jiang et al., 2011; Wan et al., 2022).
- No justification was made for the selection of ZEA exposure doses administered via feed. Referring to other authors of scientific studies is not enough
-- Thanks for the reviewer’s comments. Zearalenone levels used in the present study were based on our investigations in Shandong province of China, A 5-year survey program initiated by our research group in order to evaluate the incidence of zearalenone in feed and feed raw materials in Shandong Province of China. A total of 1 389 analyses were performed by the Asia Mycotoxin Analysis Center (Chaoyang University of Technology, Taichung, Taiwan). Results showed that the positive detection rate of ZEA reached 69.15%, the highest value of ZEA in compound feed samples was 4333.03 μg/kg and the average value of ZEA in compound feed samples was 972.56 μg·kg-1 [Chen X. X., C. W. Yang, L. B. Huang, Q. S. Niu, S. Z. Jiang* and F. Chi. Zearalenone altered the serum hormones, morphologic and apoptotic measurements of genital organs in post-weaning gilts. Asian-Australasian Journal of Animal Sciences, 2015, 28(2): 171-179.].
Indeed, 1 to 1.5 mg/kg ZEA in pigs feed is not commonly found but may happen occasionally (Zinedine et al., 2007). Chinese standard of ZEA (2006) in weaning gilts is 0.5 mg/kg, so 0.5 mg/kg, 1 mg/kg and 1.5 mg/kg were used to investigate the effects on the the location and expression of estrogen receptor ERα/β and progesterone receptor (PR) of uterus in weaned piglets. And the porcine endometrial epithelial cells (PECs) were incubated for 24 h with ZEA at 0 (Control), 5 (ZEA5), 20 (ZEA20) and 80 (ZEA80) μmol/L to reveal the mechanism underneath. Our results strongly suggested that ZEA may promote the development of uterus in weaned gilts through estrogen receptor pathway.
The newly revised Chinese standard of ZEA (2017) in weaning gilts is 0.15 mg/kg, and our study on ZEA with lower dose of 0.1-0.5 mg/kg is in progress.
- There is also no rationale for selecting ZEA doses during the 24-hour culture of porcine endometrial cells.
-- The PECs were incubated with ZEA for 24 h, under the different concentration (0、5、20、40、80、160 μmol ZEA/L). The results showed that it had no significant effect on cells under the condition of 5 μmol ZEA/L. The cell morphology changed from normal paving stone shape to irregular shape at 20 μmol ZEA/L. And then, cells were severely damaged and began to die in large numbers at 80 μmol ZEA/L. When the concentration of ZEA reached 160 μmol/L, a large number of cells shrank and floated, and their viability was very low. At the same dose of ZEA, the PECs proliferation rate at 48 hours was lower than that at 24 hours. And the proliferation rate of ZEA80 was 0.65 at 24 h and 0.48 at 48 h (less than 50%). Therefore, considering the time and dose effects, we finally chose to incubate with ZEA (0、5、20、80 μmol/L) for 24 hours. (More details are in the manuscript)
- There is no justification as to whether there is any correlation between the ZEA exposure doses (contaminants) in the feed and the exposure values during culturing the endometrial epithelial cells of gilts.
-- Since there was little literature showing the effects of ZEA on porcine endometrial epithelial cells. We comprehensively considered the exposure concentration of ZEA to different cells, such as porcine ovarian granulosa cells, oocytes, mice and bovine endometrial cells, and conducted ZEA time and dose gradient tests to determine the optimal concentration (Tiemann et al., 2003; Zhang et al., 2018; Yang et al., 2018; Zhang et al., 2018). At the same time, we judge the action concentration of ZEA according to the cell activity, and verify the in vivo results by simulating the in vivo state.
Tiemann U, Viergutz T, Jonas L, Schneider F. Influence of the mycotoxins alpha- and beta-zearalenol and deoxynivalenol on the cell cycle of cultured porcine endometrial cells. Reprod Toxicol. 2003 Mar-Apr;17(2):209-18.
Zhang RQ, Sun XF, Wu RY, Cheng SF, Zhang GL, Zhai QY, Liu XL, Zhao Y, Shen W, Li L. Zearalenone exposure elevated the expression of tumorigenesis genes in mouse ovarian granulosa cells. Toxicol Appl Pharmacol. 2018 Oct 1;356:191-203.
Yang D, Jiang X, Sun J, Li X, Li X, Jiao R, Peng Z, Li Y, Bai W. Toxic effects of zearalenone on gametogenesis and embryonic development: A molecular point of review. Food Chem Toxicol. 2018 Sep;119:24-30.
Zhang K, Tan X, Li Y, Liang G, Ning Z, Ma Y, Li Y. Transcriptional profiling analysis of Zearalenone-induced inhibition proliferation on mouse thymic epithelial cell line 1. Ecotoxicol Environ Saf. 2018 May 30;153:135-141.
- The biotransformation processes of zearalenone at the intestinal level and in the blood of the gilts during the 35-day intoxication period were completely forgotten. About adaptive processes, especially adaptive immunity, which begins around the 7th day of exposure to ZEA and its metabolites in the organism of matured gilts. The biological activity of alpha-zearalenol is much greater than that of the parent substance.
-- Thanks for your suggestions. The biotransformation processes of zearalenone at the intestinal level and in the blood of the gilts during the 35-day intoxication period were not presented in this paper, as we might have neglected this process of breeding. However, the proliferation rate of peripheral blood lymphocytes in ZEA1.0 and ZEA1.5 groups decreased significantly, which indicated that high dose of ZEA could reduce the immunity of piglets. At the same time, under the treatment of ZEA, the antioxidant capacity of duodenum, jejunum and ileum was significantly decreased, and the intestinal oxidative stress was enhanced (Chen et al., 2019, 2020, 2021). (More details are in the manuscript)
- There is no compatibility between the title, purpose and conclusions.
-- Apparent digestibility is not reflected in the title, it's just a phenotypic indicator, which would be too long to show in the title. And the title was changed to “Zearalenone promotes uterine development of weaned gilts by interfering with serum hormones and up regulating expression of estrogen and progesterone receptors”.
-- L48-50 (old) [L55-59 (new)]: The sentence “The main objective of this study was to investigate the effects of feed supple-mented ZEA at different concentrations on serum hormones of weaned piglets and the expressions of ERs and PR of porcine endometrial epithelial cells” was changed to “Therefore, the main objective of this study was to investigate the effects of different concentrations of ZEA on serum reproductive hormones and hormone receptors (es-trogen and progesterone receptors) of weaned gilts, and to conduct in vitro verification through porcine endometrial epithelial cells, so as to explore the potential mechanism of ZEA in promoting uterine development.”
- There is a provision for uterine development in the abstract and conclusion. In my opinion, there is degeneration and atrophy of celles and tissues, particularly in endometrium and myometrium and extensive edema and numerous extravasations were typical of estrogen-induced lesions. Therefore, it is not development, but a disease state (pathology) accompanying zearalenone mycotoxicosis.
-- We have confirmed that ZEA promoted uterine development from multiple perspectives such as up‐regulating growth hormone receptor expression (Zhou et al., 2019), activating the TGF-β1/Smad3 signaling pathway (See the figure below, Zhou et al., 2018). The extensive edema and numerous extravasations ZEA-induced need further study to confirm. (More details are in the manuscript)
Detailed comments:
- Lines 40-43 - The interaction of ZEA with steroid hormones, ERs, PRs and the hypothalamic-pituitary-gonad axis has been forgotten;
-- L44-45 (new): The sentences “HPG axis plays an important role in reproductive endocrine function. It was demonstrated that endocrine disruptors could disturb HPG axis, and it adversely affected the function of the reproductive system. Nevertheless, the estrogenic compound has the potency to bind with the nuclear ER throughout the HPG axis and impairs the reproductive function by altering steroid hormone synthesis, such as estrogen and progesterone. Meanwhile, the variation of reproductive hormones might affect the HPG positive and negative feedback loop.” were added.
- Line 155 - What does purity mean - were the presence of ZAE metabolites checked, for example?
-- The word “purity” means that the purity of ZEA added reached 98%.
- Line 155 - I think there should be another term. Commensalism is a long-term biological interaction (symbiosis) in which members of one species gain benefits while those of the other species neither benefit nor are harmed. It is an undesirable substance, i.e. a substance that is present in the environment without human intervention but that may be dangerous to mammals.
-- I didn’t find the commensalism you mentioned in Line 155. If possible, please help me clarify the specific location
- Lines 183-185 - It is difficult to relate the results of pregnant sows to the results of maturing gilts
-- Yes, it is difficult to relate the results of pregnant sows to the results of maturing gilts. Therefore, the sentence “Studies have showed that ZEA has a time-dependent effect on serum hormones, although serum LH concentration was increased in ZEA group (0.25 mg/kg) during middle gestation and after farrowing” was deleted.

Reviewer 2 Report
Dear authors,
Thanks for submitting this interesting paper. I think the study is valuable and should be published with minor revisions. I have tried to sum up my comments and questions below:
Introduction: The statement that only 25% of the crops worldwide is contaminated by mycotoxins was recently disproved by the following publication: https://pubmed.ncbi.nlm.nih.gov/31478403/.
I have corrected some spelling error within the pdf file seen as comments – please correct them.
General questions:
· Did you investigate the size and/or weight of uterus and ovaries? At these high concentrations, usually an enlargement of the organs can be seen. Furthermore, did you see clinical effects such as vulva reddening?
· Did you check the feed also for other mycotoxins such as alternariol, which is described to be estrogenic as well? Or also fumonisins and deoxynivalenol?
· And you should analyse the feed also for phytoestrogens such as daidzein and genistein, which might also contribute to an estrogenic effect. Is it possible to re-analyse the feed for those components?
· Why did you choose such high concentration of ZEA (0.5 – 1.5 mg/kg? In Europe, the guideline for ZEA in piglets is to not exceed 100 ppb (µg/mL).
Best regards,
Reviewer
Author Response
Dear reviewer 2:
On behalf of my co-authors, we thank you very much for giving us an opportunity to revise our manuscript (toxins-1947616), we appreciate you very much for their positive and constructive comments and suggestions on our manuscript. Those comments are all valuable and very helpful for revising and improving our researches. We have studied the valuable comments carefully and have made correction which we hope meet with approval.
- Introduction: The statement that only 25% of the crops worldwide is contaminated by mycotoxins was recently disproved by the following publication: https://pubmed.ncbi.nlm.nih.gov/31478403/.
-- L25-27 (old) [L26-28 (new)]: The sentence “According to United Nations Food and Agriculture Organization, approximately 25% of food and feed output worldwide were contaminated by mycotoxins, the most important of which is Fusarium mycotoxins” was changed to “According to United Nations Food and Agriculture Organization, approximately 25% of food and feed output worldwide were contaminated by mycotoxins [1], even the prevalence for the detected mycotoxins is up to 60–80%, according recently research [2]”.
- Did you investigate the size and/or weight of uterus and ovaries? At these high concentrations, usually an enlargement of the organs can be seen. Furthermore, did you see clinical effects such as vulva reddening?
-- The uterus of piglets in the ZEA treatment group was significantly larger than that in the control group. And the thickness of myometrium and endometrium increased with higher level of ZEA. Those results were shown in Zhou et al. (2018).
- Did you check the feed also for other mycotoxins such as alternariol, which is described to be estrogenic as well? Or also fumonisins and deoxynivalenol?
-- The levels of vomittin, aflatoxin and fumigatoxin in the diet were entrusted to Qingdao Entry Exit Inspection and Quarantine Bureau for determination, and the contents were lower than the minimum detection standards.
- And you should analyse the feed also for phytoestrogens such as daidzein and genistein, which might also contribute to an estrogenic effect. Is it possible to re-analyse the feed for those components?
-- We didn’t analyse the feed for phytoestrogens such as daidzein and genistein, since the plant components, especially soybean, in the diets of each treatment group were the same.
- Why did you choose such high concentration of ZEA (0.5 – 1.5 mg/kg? In Europe, the guideline for ZEA in piglets is to not exceed 100 ppb (µg/mL).
-- Thanks for the reviewer’s comments. Zearalenone levels used in the present study were based on our investigations in Shandong province of China, A 5-year survey program initiated by our research group in order to evaluate the incidence of zearalenone in feed and feed raw materials in Shandong Province of China. A total of 1 389 analyses were performed by the Asia Mycotoxin Analysis Center (Chaoyang University of Technology, Taichung, Taiwan). Results showed that the positive detection rate of ZEA reached 69.15%, the highest value of ZEA in compound feed samples was 4333.03 μg/kg and the average value of ZEA in compound feed samples was 972.56 μg·kg-1 [Chen X. X., C. W. Yang, L. B. Huang, Q. S. Niu, S. Z. Jiang* and F. Chi. Zearalenone altered the serum hormones, morphologic and apoptotic measurements of genital organs in post-weaning gilts. Asian-Australasian Journal of Animal Sciences, 2015, 28(2): 171-179.].
Indeed, 1 to 1.5 mg/kg ZEA in pigs feed is not commonly found but may happen occasionally (Zinedine et al., 2007). Chinese standard of ZEA (2006) in weaning gilts is 0.5 mg/kg, so 0.5 mg/kg, 1 mg/kg and 1.5 mg/kg were used to investigate the effects on the the location and expression of estrogen receptor ERα/β and progesterone receptor (PR) of uterus in weaned piglets. And the porcine endometrial epithelial cells (PECs) were incubated for 24 h with ZEA at 0 (Control), 5 (ZEA5), 20 (ZEA20) and 80 (ZEA80) μmol/L to reveal the mechanism underneath. Our results strongly suggested that ZEA may promote the development of uterus in weaned gilts through estrogen receptor pathway
The newly revised Chinese standard of ZEA (2017) in weaning gilts is 0.15 mg/kg, and our study on ZEA with lower dose of 0.1-0.5 mg/kg is in progress.

Reviewer 3 Report
The submitted work is of adequate quality for publication in this journal and I have only mild comments about it. Keywords are repeated in the title.
In the material and methodology section, please state the permission of the experiment by the ethics committee. Also state how the pigs were euthanized.
In the table number 3 is not listed correctly tryptophan.
Author Response
Dear reviewer 3:
On behalf of my co-authors, we thank you very much for giving us an opportunity to revise our manuscript (toxins-1947616), we appreciate you very much for their positive and constructive comments and suggestions on our manuscript. Those comments are all valuable and very helpful for revising and improving our researches. We have studied the valuable comments carefully and have made correction which we hope meet with approval.
- In the material and methodology section, please state the permission of the experiment by the ethics committee. Also state how the pigs were euthanized.
-- L263-266: The piglets in the experiment were raised in accordance with the guidelines for the care and use of laboratory animals provided by the Animals Nutrition Research Institute of Shandong Agricultural University and the Ministry of Agriculture of China (SDAUA‐2020‐0710).
-- L323-324: The sentence “All piglets were euthanized immediately after blood collections and two uterine samples were immediately excised sterilely” was changed to “All piglets were euthanized by electrocution (head only, 110 V, 60 Hz) after blood collections and two uterine samples were immediately excised sterilely”.
- In the table number 3 is not listed correctly tryptophan.
-- The content of tryptophan was 0.22%, which was shown in Table 3.
Round 2
Reviewer 1 Report
It would be good to explain what HPG is.
Record mg ZEA / kg of feed.
I expected that the responses tp my comments or suggestions would be introduced to the paper, not only for my message.
Author Response
Dear reviewer 1:
On behalf of my co-authors, we thank you very much for giving us an opportunity to revise our manuscript (toxins-1947616), we appreciate you very much for their positive and constructive comments and suggestions on our manuscript.
According to your suggestion, we present the relevant content in the paper (not only in the message), so as to make the article more logical and conducive to the reader's understanding.
- It would be good to explain what HPG is.
L45-48: Hypothalamic-pituitary-gonadal (HPG) axis plays an important role in reproductive endocrine function. It was demonstrated that endocrine disruptors could disturb HPG axis, and it adversely affected the function of the reproductive system.
- Record mg ZEA / kg of feed.
According to your suggestion, the words “ mg/kg” in this paper has been changed to “mg ZEA / kg”.
